# The Role of Different Types of microRNA in the Pathogenesis of Breast and Prostate Cancer

**DOI:** 10.3390/ijms24031980

**Published:** 2023-01-19

**Authors:** Ekaterina A. Sidorova, Yury V. Zhernov, Marina A. Antsupova, Kamilya R. Khadzhieva, Angelina A. Izmailova, Denis A. Kraskevich, Elena V. Belova, Anton A. Simanovsky, Denis V. Shcherbakov, Nadezhda N. Zabroda, Oleg V. Mitrokhin

**Affiliations:** 1Department of General Hygiene, F. Erismann Institute of Public Health, I.M. Sechenov First Moscow State Medical University (Sechenov University), 119435 Moscow, Russia; 2Department of Chemistry, Lomonosov Moscow State University, 119991 Moscow, Russia; 3Center of Life Sciences, Skolkovo Institute of Science and Technology, 121205 Moscow, Russia; 4Center for Medical Anthropology, N.N. Miklukho-Maclay Institute of Ethnology and Anthropology, Russian Academy of Sciences, 119017 Moscow, Russia

**Keywords:** breast cancer, prostate cancer, microRNA, 3P medicine, non-coding RNA, tumor biomarkers, metastatic disease, molecular mechanisms

## Abstract

Micro ribonucleic acids (microRNAs or miRNAs) form a distinct subtype of non-coding RNA and are widely recognized as one of the most significant gene expression regulators in mammalian cells. Mechanistically, the regulation occurs through microRNA binding with its response elements in the 3’-untranslated region of target messenger RNAs (mRNAs), resulting in the post-transcriptional silencing of genes, expressing target mRNAs. Compared to small interfering RNAs, microRNAs have more complex regulatory patterns, making them suitable for fine-tuning gene expressions in different tissues. Dysregulation of microRNAs is well known as one of the causative factors in malignant cell growth. Today, there are numerous data points regarding microRNAs in different cancer transcriptomes, the specificity of microRNA expression changes in various tissues, and the predictive value of specific microRNAs as cancer biomarkers. Breast cancer (BCa) is the most common cancer in women worldwide and seriously impairs patients’ physical health. Its incidence has been predicted to rise further. Mounting evidence indicates that microRNAs play key roles in tumorigenesis and development. Prostate cancer (PCa) is one of the most commonly diagnosed cancers in men. Different microRNAs play an important role in PCa. Early diagnosis of BCa and PCa using microRNAs is very useful for improving individual outcomes in the framework of predictive, preventive, and personalized (3P) medicine, thereby reducing the economic burden. This article reviews the roles of different types of microRNA in BCa and PCa progression.

## 1. Introduction

According to the Global Cancer Observatory’s (GLOBOCAN) estimates for 2020, breast cancer (BCa) is the most common cancer among women worldwide and accounts for nearly 2.3 million reported cases [1]. The five-year survival rate is less than 30%, despite all the successes of modern treatment approaches. Median incidence is 61 years [1]. Prostate cancer (PCa) is a commonly diagnosed disease worldwide. It has amounted 1,414,249 new cases of PCa and 375,000 deaths from this disease in 2020 [1]. In a third of patients with PCa, the tumor progresses (perineural and stromal invasion) after initial regression in response to androgen deprivation therapy [2]. Despite modern advances in surgical, chemotherapeutic, and radiation treatments, the five-year survival rate in castration-resistant patients is approximately 31.0% [3]. Most of the deaths associated with PCa are due to the failure of existing treatments to prevent tumor spread [4].

This rate of increase in the incidence of BCa and PCa compels the use of the latest screening programs and strengthens the need for preventive measures aimed at reducing the incidence of the disease. Thus, primary, secondary, and tertiary care, which is provided within the framework of predictive, preventive, and personalized (3P) medicine, is a promising strategy for the development of new approaches in therapy and shows excellent results both in terms of individual results and economic costs. A paradigm shift from reactive treatment of symptomatic PCa to a prognostic approach and personalized prevention is essential [5,6]. 

One of the latest developments is the detection of microRNAs. These molecules have been shown to contribute significantly to carcinogenesis, disease progression, and the acquisition of resistance to traditional antitumor drugs.

Micro ribonucleic acids (microRNAs or miRNAs) were isolated in the early 1990s from *Caenorhabditis elegans* [7]. They are small (18–25 nucleotides) non-encoded single-stranded molecules whose main function is to regulate the expression of a particular gene at the transcription stage and in the post-transcriptional period. Compared to small interfering RNAs (siRNAs), microRNAs have more complex regulatory patterns, making them suitable for fine-tuning gene expression in different tissues (Table 1). A microRNA binds to the non-coding part of the target RNA of the desired gene and manifests its functions through silencing systems, preventing the above steps of gene expression, and can also inhibit biological processes, such as cell differentiation, proliferation and development, depending on the nucleotide sequence [8,9].

An interesting aspect is that the same microRNA in different systems can have opposite functions, depending on the pathological or physiological conditions, or the kind of connections and with which genes the connections are established. According to other data, the predominant number of targets in different tissues does not differ much, but different isoforms of the 3’-untranslated region (3’-UTR) and landscape may affect the resulting functions [7].

The regulation of gene expression is of great importance as part of the cellular response to negative environmental influences, such as hypoxia, starvation, and DNA damage [10]. MicroRNA acts as a control unit for the work of proteins responsible for DNA repair, therefore microRNAs can function as oncogenes (oncomirs) or be tumor growth suppressors (oncosuppressors) [11].

Different types of cancer may share aberrant microRNAs, although most microRNAs are specific to each type of cancer. Thus, there are a number of specific microRNAs for BCa, but they differ in their mechanism of action and role in the development of BCa. Differential microRNA expression is strongly associated with specific tumor stages, lymph node metastases, poor prognosis, and response to specific therapies [7]. 

Prostate carcinomas show biodiversity and can either be localized to the prostate or become highly invasive and metastasize to regional lymph nodes and other organs [12]. Although localized prostate carcinoma is successfully treated, treatment efficacy is greatly reduced when prostate tumor cells metastasize outside the gland, mainly through perineural and stromal invasion. The process of metastasis is diverse, but the mechanisms that contribute to the spread of the metastatic phenotype in prostate carcinomas are not well understood at the molecular level. The epithelial-mesenchymal transition (EMT) is a key process for the transition from non-invasive to invasive PCa, in which polarized epithelial cells lose their tight intercellular junctions, which leads to an increase in their migratory ability, an increase in invasive properties, and the acquisition of a mesenchymal phenotype (Figure 1) [13].

MicroRNAs (microRNAs) are involved in the regulation of up to 60% of the protein-coding genes [14]; correspondingly, assessment of microRNA expression in various cancer diseases is a topical problem in molecular oncology in recent years. A considerable part of such studies is focused on the search for tumor-specific microRNAs potentially utile in designing the diagnostic systems for cancer diseases [15].

MicroRNAs form a distinct non-coding RNA subtype and are widely recognized as one of the important regulators of gene expression in mammalian cells [16].

Mechanical regulation occurs through the binding of microRNAs to microRNA response elements (MREs) in the 3’-untranslated regions (3’-UTRs) of target messenger RNAs (mRNAs) [17]. This leads to the post-transcriptional silencing of the genes expressing these mRNAs [17]. The mode of silencing in the RNA-induced silencing complex (RISC) depends on the complementarity between the MRE mRNA and the guide microRNA [18]. If the molecules are completely or nearly perfectly complementary, RISC cleaves the target mRNA [18]. If microRNA is not fully complementary to mRNA, then RISC can bind to the MRE sequence only by the heptameric 5’-seed region of the guide microRNA, accelerating mRNA decay due to deadenylation or preventing translation at its various stages by blocking initiation factors and binding of ribosome subunits, slowing down elongation or contributing to premature termination [18]. Thus, microRNAs, compared to small interfering RNAs, have more complex regulatory patterns that are suitable for fine-tuning gene expressions in different tissues (Figure 2).

The role of microRNAs as one of the main causal factors in the growth of malignant cells was first proposed in 2002 [19]. Currently, there is more and more data on the dysregulation of microRNAs in the transcriptomes of various types of cancer. MicroRNA expression profiles change during the development of most malignant tumors, which suggests that microRNAs can act as oncogenes, tumor suppressors, or drivers of malignant transformation [20]. There is also growing evidence of microRNA dysregulation in PCa. Blood patterns play one of the key role in prognostic diagnosis, targeted prevention, and the development of personalized treatment algorithms [21,22].

The importance of the early diagnosis of prostate and breast cancers is determined by the high aggressiveness of the disease development. Diagnostics of risk of these cancer types gives the patient the opportunity to choose between active surveillance and tumor surgical resection, which is an appropriate therapeutic approach, greatly enhancing the survival of the patient. Much more work is needed to develop robust diagnostics for PCa screening in the future [23].

## 2. The Role microRNA in Prostate and Breast Cancer

MicroRNA-141 was first mentioned as a potential diagnostic marker in 2008 [24]. The authors found that miR-141 was overexpressed 46-fold in a patient sample and could distinguish between advanced cases of metastatic PCa and healthy individuals with a specificity of 60% and a sensitivity of 100% [24].

MicroRNA-141, microRNA-375, and microRNA-200b showed the highest correlation with clinical parameters of PCa. Also, miR-375 expression showed the highest correlation with tumor stage, Gleason score, and has a considerable association with lymph-node metastasis [25].

The potential of microRNA-375 was found as prognostic biomarker of PCa by reported significant association of high expression of microRNA-375 levels with shorter overall survival. Therefore, microRNA-375 expression correlates with clinicopathological parameters and can be a prognostic biomarker of PCa [26].

Another study showed that the expression of microRNA-141 together with microRNA-375 helps to differentiate patients with metastatic PCa and patients with low-risk localized cancer [27].

The expression of microRNA-153 in BCa tissue samples and 128 MDA-MB-231 cells was found to be significantly lower than normal. This group of scientists has shown that microRNA-153 inhibits migration, invasion, and epithelial-mesenchymal transition of BCa by regulating the signaling pathway of transforming growth factor beta (TGF-β) (Figure 3) [28].

Increased expression of microRNA-153 in PCa was discovered in 2013. It is known that microRNA-153 plays an important role in stimulating the proliferation of human PCa cells and represents a new mechanism for direct suppression of phosphatase and tensin homolog (PTEN) expression in PCa cells mediated through microRNA [29]. 

MicroRNA-153 is up-regulated in PCa tissues and may play a major role in aggressive PCa by attacking potential target genes [30].

So far, data on the clinical significance of microRNA-153 expression in PCa are scarce. Thus, it was found that the high expression of microRNA-153 in PCa tissues closely correlated with aggressive clinical pathological parameters, such as metastasis to the lymph nodes, to the bones, a high level on the Gleason score, and a high stage on the TNM Classification of Malignant Tumors (TNM). PCa patients with high levels of microRNA-153 expression had a clearly lower 5-year overall survival compared to patients with low microRNA-153 expression. Importantly, Cox’s multivariate regression analysis showed that microRNA-153 expression was an independent predictor of 5-year overall survival in patients with PCa [31].

## 3. Oncomirs and Oncosuppressive microRNAs

### 3.1. Oncomirs

One group of oncomirs are microRNAs that are involved in metastasis, including miR-21, miR-181-a, miR-632, and miR-221/222 [7]. Tumor cells form clusters for metastasis; the tumor microenvironment, composed of M2-like macrophages (stromal, tumor, endothelial, and infiltrating immune cells) plays a critical role in the process of metastasis [32]. The most studied microRNA from this group is microRNA-21; its target genes are tropomyosin 1 (TPM1), programmed cell death 4 (PDCD4), and metallopeptidase inhibitor 3 (TIMP3). Increased expression of this oncomir usually indicates a poor prognosis of the course of the disease [7]. 

A second group of oncomirs, including miR-155 and miR-375, are involved in the proliferation of tumor cells [7]. MiR-155 is multifunctional: it is responsible for proliferation, migration, drug resistance, and immune evasion by reducing apoptotic activity. The target genes of this oncomir are suppressor of cytokine signaling 1 (SOCS1) and signal transducer and activator of transcription 3 (STAT3). Some recent publications also mention the role of miR-155 in the metabolism of thiamine, glucose, and estrogen [33].

A third group, represented by miR-182, miR-10b, miR-373, and miR-520c, participate in the invasion of tumor cells into tissues. MiR-10b is also involved in BCa metastasis through the twist-related protein 1 (TWIST1) target gene, and in the invasion and migration of BCa cells through the homeobox D10 (HOXD10) and T cell lymphoma invasion and metastasis-inducing protein 1 (Tiam1) target genes. MiR-10b expression level positively correlates with clinical stage and lymph node metastases [7].

The next group includes two oncomirs, miR-9 and miR-27a, which are involved in tissue vascularization or angiogenesis. It was suggested that miR-9, through the target genes E-cadherin and vascular endothelial growth factors (VEGF), can favorably influence the invasiveness, metastasis, and neoangiogenesis of BCa [34].

The last group includes miR-22, miR-181-a, and miR-221/222. These oncomirs are predominantly involved in the epithelial-mesenchymal transition [7]. Due to their multifunctionality, miR-221/222 were included in two groups; these oncomirs play the most important role in the regulation of genes involved in adhesive junctions, signaling in the phosphoinositide 3-kinases (PI3K) and mitogen-activated protein kinase (MAPK) pathways. These oncomirs are considered prometastatic and proangiogenic (Figure 4) [32].

### 3.2. Oncosupressors

Suppressor microRNAs are also divided into groups depending on the processes of tumor development that they are involved in: initiation, apoptosis, proliferation, tumor growth, invasion, angiogenesis, or epithelial-mesenchymal transition.

One of the main oncosuppressors is the let-7 family. Overexpression of let-7 in BCa at the initial stages inhibits the activity of Harvey rat sarcoma viral oncogene homolog (H-RAS) and High-mobility group AT-hook 2 (HGMA2) [35]. Let-7b expression is often downregulated in lymph node metastases. This microRNA inhibits cell migration and invasion through the following target genes: serine/threonine-protein kinase (PAK1), diaphanous-related formin 2 (DIAPH2), radixin (RDX), and integrin subunit beta (ITGB) [7].

MiR-145 is also one of the most studied suppressor microRNAs. It is known that miR-145 expression is reduced in cancer cells compared to normal tissues [36]. Mir-145 is a regulator of the MUC1 membrane protein and also inhibits angiogenesis and tumor growth by suppressing the neuroblastoma RAS viral oncogene homolog (N-RAS) and VEGF [32,36]. It is also known that this oncosuppressor is underexpressed in patients with a tumor larger than 2 cm and metastases in the lymphatic system compared with patients without metastases in the nodes and with smaller tumors [36]. 

There are ambiguous data on the role of miR-205 in the development of BCa, but still, most studies indicate the oncosuppressive function of this microRNA through the effect on the zinc finger E-box binding homeobox 1 and 2 (ZEB1, ZEB2) genes, which are the initiators of EMT [9]. The higher the expression of miR-205 in tumor cells the better, since a decrease in its expression usually indicates the progression of BCa, leading to a rapid deterioration in the prognosis and possible metastasis to the lymph nodes [37]. 

The miR-200 microRNA family includes miR-200a, miR-200b, miR-200c, miR-429, and miR-141. Most studies have found that miR-200 suppresses the epithelial-mesenchymal transition by regulating the expression of ZEB1/2 genes. Thus, this family plays an important role in the invasion of solid tumors (Figure 5) [38]. 

### 3.3. The Role of microRNA in Breast Cancer

BCa is a heterogeneous disease and its classification has been based on such parameters as histological characteristics (tumor size, histological degree, lymph node involvement), and the human epidermal growth factor receptor (HER2) status in conjunction with the analysis of the overall patient condition, body reactivity, and age [39]. Advances in molecular techniques during the last decades have allowed scientists to classify BCa basing on immunohistochemistry (ICH) analysis. This complex bioinformatic analysis allows for the identification of well-known biomarkers of BCa: estrogen receptor (ER), progesterone receptor (PR), HER2, and Ki67 (proliferation index marker). In recent decades, four molecular subtypes of BCa have been revealed based on the understanding of their genetic profile: luminal A, luminal B, HER2-enriched, and basal-like (triple negative). The treatment strategies, prognoses, and survival rates are different for each of them. In one study, 186 samples were sequenced for microRNA expression. An average of 684 different microRNAs were observed in one sample, the most common of which was miR-21-5p in all tumor samples. In this study all specimens were divided into three clusters based on the expression patterns of ER, PR, HER2, and Ki67. The luminal A subtype was assigned to Cluster 1 along with several specimens of the luminal B subtype. The analysis revealed that ER+ tumors expressed the following microRNAs: miR-26, miR-5681a, miR-5695, miR-887, miR-149, miR-375, miR-342, miR-190b, miR-29c, miR-29b, and miR-499a. Separately, miR-99a/let-7c/miR-125b-2 expression was shown for the luminal A subtype. For tumors with high HER2 expression, the correlation with miR-4728 was shown. MiR-4728 plays a dual role in these tumors, on the one hand as the suppressor gene, and on the other hand as reducing anti-HER2 therapy activity [40,41,42] (Table 2).

### 3.4. The Role of miR-99a

Patients with miR-99a expression have better survival rates, which correlates with the belief that the luminal A subtype has good survival rates in general [40]. It is shown that miR-99a regulates multiple signaling pathways and initiates the change of the entire transcriptome [43].

One of the targets for miR-99a is FGFR3 (fibroblast growth factor receptor 3). The gene of this receptor is located on chromosome 4p16.3, which encodes a protein whose extracellular part can interact with fibroblast growth factors and initiate a cascade of downstream signals that regulate mitogenesis and differentiation. Overexpression of this receptor can trigger a cascade of reactions that are key mediators of malignant tumors, for instance, PI3K-AKT, RAS/RAF/MEK/MAPK. MiR-99a directly targets FGFR3. High expression of miR-99a decreased FGFR3 expression and vice versa (Figure 6) [43].

The role of miR-99a was also shown in relation to mTOR, which is associated with the PI3P/AKT pathway, although it can be activated through others. The targets of mTOR are 4E-BP1 and S6K1 proteins, which stimulate tumor cells to proliferate, increasing transcription. MiR-99a significantly reduces mTOR expression by affecting the mRNA and proteins. MiR-99a inhibits the phosphorylated form of mTOR below the p-4E-BP1 and p-S6K1. In addition, MiR-99a is statistically inversely correlated with mTOR expression in breast tissue samples and cell lines [44].

The correlation between miR-99a and expression of insulin-like growth factor 1 (IGF-1) has also been studied. IGF-1 is a transmembrane protein belonging to the large tyrosine kinase group [38]. IGF-1R and its ligands may contribute to human cancer progression. Overexpression of IGF-1R can activate various downstream signaling pathways, such as PI3K/AKT, MAPK/ERK, EP2/EP4, and RAS/RAF/ERK, which play a key role in the propagation and apoptosis of malignant tumors. Furthermore, IGF-1R overexpression may affect the progression of the epithelial-mesenchymal transition (EMT). MiR-99a overexpression can markedly inhibit the proliferation of tumor cells, migration and invasion in vitro, and reduce the growth of tumor implants in vivo. MiR-99a can bind to 3’-UTR IGF-1R, thereby reducing the activity of this protein by disrupting the transcription [45,46].

Another important target for miR-99a is HOXA1. The homeobox family (HOX) encodes homeodomain-containing transcription factors that are involved in various physiological processes that include embryonic development, cell proliferation, and differentiation. HOXA1 is involved in tumor progression and prediction of certain types of human cancer. Overexpression of HOXA1 was associated with poor prognosis and advanced clinical pathological features in patients with BCa. Additionally, knockdown HOXA1 significantly inhibited cell proliferation by increasing cell apoptosis and stopping cell cycles, which was accompanied by the aberrant expression of cell cycle and apoptosis-related proteins, cyclin D1, B cell lymphoma 2 (BCL-2), and BCL-2 protein 4 [47]. High expression of miR-99a reduces endogen HOXA1 protein. The use of three algorithms (TargetScan, PicTar, and miRanda) for miR-99a target prediction first demonstrated that HOXA1 is a direct functional target of miR-99a for BCa. MiR-99a and HOXA1 have an inverse dependence of expression, so when the expression of one increases, the expression of the other decreases [48].

### 3.5. The Role of miR-21-5p

miR-21-5p is the most common microRNA that can express in BCa. In recent years, it has become a trend to think that miR-21-5p might be the key predictor biomarker for BCa. MiR-21-5p is connected to several main cascade signaling pathways [49]. For example, miR-21-5p enhances tumor progression by targeting genes, such as leucine zipper transcription factor like 1 (LZTFL1), programmed cell death 4 (PDCD4), and tropomyosin 1 (TPM1) [49].

Leucine zipper transcription factor like 1 (LZTFL1) is a gene located on chromosome 3p21.3, which is a tumor suppressor gene (TSG) associated with the signaling pathway that involves E-cadherin (E-cad) and activates EMT (Figure 7) [50,51,52].

E-cad is one of the biological markers of epithelial cells, responsible for stability of cell adhesion, polarity, and signal transmission [53]. EMT associated with loss of E-cad expression contributes to the cells losing their normal structure and acquiring the ability to penetrate across the basement membrane, and thus provides for long distance metastases [52,53,54].

Interestingly, partial EMT cells are required for the rapid metastasis formation of nearby tissues and organs. The partial EMT cells most frequently metastasize into lungs and contribute to chemoresistance, while cells with full EMT have a lower metastatic ability and cannot colonize the lungs [54].

MiR-21-5p down-regulates the LZTFL1 gene, thus preventing this gene from acting directly on E-cad, thus increasing the EMT-related processes. Without the inhibitory action of miR-21-5p, the LZTFL1 gene is able to increase the expression of E-cad in tumor cells when it begins to decrease by suppressing the β-catenin nuclear translocation, which oppositely increases miR-21-5p [51].

Programmed cell death 4 (PDCD4) is a tumor suppressor gene involved in the apoptosis of malignant cells, which is located in the 10q24 chromosome. MiR-21-5p directly binds to 3-UTR PDCD4 and thus inhibits this gene and all the related processes that can lead to bad predictions for patients [55]. Normally (for normal expression in cancer cells), PDCD4 inhibits the interaction between EIF4A1 and EIF4G, which reduces the helicase activity of EIF4A1, thereby disrupting translation and resulting in impaired growth factor formation, growth stimulation genes, and proto-oncogenes [50,56].

Overexpression of PDCD4 reduces proliferation and activates apoptosis in the following BCa cell lines: T-47D, MDA-MB-231, and MCF-7. Reduced PDCD4 expression in HER2− and ER+ promotes metastasis to lymph nodes and invasions [56]. Moreover, ER+ are in close correlation with miR-21-5p. ER+ can enhance the suppressive action of miR-21-5p on PDCD4. Interestingly, in hormonal ER+ therapy of positive tumors, PDCD4 increases its expression, thereby contributing to the effectiveness of the treatment. Thus, suppressing PDCD4 ensures the formation of a tolerance to the treatment [56,57].

In HER2+ tumors, PDCD4 expression increases when treated by their antagonists, which increases the effectiveness of treatments and reduces the progression [58]. MiR-21-5p is able to increase HER2 expression via STAT3, which reduces PDCD4 expression [56].

Thus, in HER2+ breast tumors, HER2 promotes progression when it changes its transcription and amplification, as it mediates the transduction signal by heterodimer and the autophosphorylation of tyrosine kinase, resulting in the activation of key cascade pathways Ras/MAPK and PI3K/Akt.

### 3.6. The Role of miR-4728

Unlike all the microRNAs described above, miR-4728 is particularly interesting for its dual role. On the one hand, it contributes to the development of resistance to therapy, in particular anti-HER2, by genetic amplification, progressing cancer, and reducing overall survival. On the other hand, miR-4728 is a tumor gene antagonist that reduces pERK activity.

There are two main targets that can be suppressed by HER2. One of them is the pro-apoptotic protein NOXA. NOXA, also known as PMAP1, is a protein responsible for activating apoptosis in human cells. It is considered one of the prognostic markers of kidney cancer, but in a recent study it was found in breast tissue for the first time [41,50].

NOXA is capable of increasing caspase activity (Figure 8), increasing the mitochondrial ability to alter the structure of its membranes. This protein can contribute to the degradation of the MCL1 gene, which encodes antiapoptotic proteins, and is associated with the protein family BCL-2 in HER2+ BCa tumors [41,50].

NOXA competes with BAK1 in binding with MCL-1, and can displace BAK1 from the binding site on MCL-1 as they have similar conformations. It is also able to displace BIM/BCL2L11 from the binding site on MCL-1 [50]. Apoptosis is one of the indicators of the effectiveness of targeted therapy. NOXA not only has pro-apoptotic effects, but also increases the sensitivity of cells to anti-HER2 drugs. By analyzing BCa cell line panels, NOXA expression is correlated with HER2 and ER expression. It has been shown that high HER2 expression reduces NOXA expression, whereas ER expression increases it [41,59]. MiR-4728 is capable of acting as a NOXA suppressor at high expressions of HER2 (Figure 9).

MiR-4728 inhibits the gene encoded by NOXA and directly reduces its expression, indirectly through the effect on mRNA ERα (ESR1), which thus acts as a transcriptional factor [41,59]. Inhibition of NOXA is not the only way that miR-4728 reduces sensitivity to therapy and thus worsens prognosis. ErbB3-binding protein 1 (EBP1) is also a target for miR-4728. EBP1 has two isoforms, p42 and p48, and only p42 can bind to ErbB3, which is known as a tumor suppressor. Its suppressive action is associated with inhibition of the p85 PI3K and AKT subunit [60].

The protein also interacts with and inhibits eukaryotic translation initiation factor 4E, which inhibits protein translation. An important influence on carcinogenesis is the suppression of growth factor translations that are involved in signaling pathways associated with MAP kinase and mTOR1. MiR-4728 can directly inhibit EBP1 and thus contribute to tumor proliferation and progression, especially in HER2+ tumors. MiR4728-5p is associated with repressive action on EBP1 [50].

Positive effects of miR-4728 are associated with the downregulation of pERK [59,61]. ERK is the key activator of ErbB2 and the signaling pathways associated with it, in particular PI3K/AKT/mTOR and MAPK. MiR-4728 blocks ERK and reduces the progression of PMP, as it reduces proliferous processes and translation in cancer cells.

## 4. Conclusions

Our literature analysis has shown that studying microRNA has become the new trend. We discovered the most common and the most studied microRNAs known for PCa and BCa. These microRNAs can differently influence tumorigenesis, treatment, and clinical outcome in patients with PCa and BCa.

MicroRNAs, such as miR-4728 and miR-21-5p, can predict bad prognosis for patients with BCa, reduce treatment efficacy, and promote tumor proliferation, invasion, and progression. The detection of miR-21-5p is important for predicting possible metastases. Its expression activates EMT, which in turn (both full and partial EMT) reduces the response to chemotherapy, with partial EMT forming metastases in the lungs, which significantly reduces the overall condition that in turn can delay the transition to further treatment, such as surgery or radiation therapy. The increased expression of miR-21-5p reduces PDD4 expression, which in turn increases the degree of invasion and metastasis of the tumor, and thus affects the patient’s survival as a result of high metastases rates. Most importantly, suppressing the expression of this molecule reduces the response to targeted therapy, especially trastuzumab-based monotherapy. Other important microRNAs known to influence BCa and PCa are group of microRNAs responsible for predicting the treatment resistance and tumor progression: miR-155 and miR-375. MicroRNAs such as miR-182, miR-10b, miR-373, and miR-520, are also responsible for the invasion and metastasis of lymph nodes. The microRNAs described above, by acting on almost the same cascade pathways, significantly affect the clinical stage of the tumor and the course of the disease. All of these microRNAs may become future therapeutic targets that will significantly improve treatment effectiveness, as we believe such an accurate approach will allow for the personalized selection of drugs for patients, which will improve predictions and survival. 

microRNA expression in PCa and BCa can also contribute to a good prognosis and outcome for the patient. miR-99a, miR-let7, miR-145, miR-200, and miR-205 are associated with suppressive action on almost all key cascade pathways that involve malignancy. These microRNAs are able to reduce the activity of angiogenesis and the formation of growth factors, which prevent the proliferation of tumor cells. This can all significantly improve patient prognosis, predict outcome, and promote personalized treatments for PCa and BCa. Effective PCa and BCa management requires a paradigm shift from reactive to 3P medicine.

We think that the diagnosis of microRNA expression has every reason to enter clinical practice because the precise examination of these molecules will allow a personalized approach to patients. The use of predictive biomarkers will be important for the optimal development of targeted therapy. Currently, the full understanding of all the mechanisms that can be controlled by different microRNAs is limited. Undoubtedly, more clinical trials need to be done, as studying microRNA can offer new treatment approaches not only for patients with PCa and BCa, but also with other types of cancer too.

## Figures and Tables

**Figure 1 ijms-24-01980-f001:**
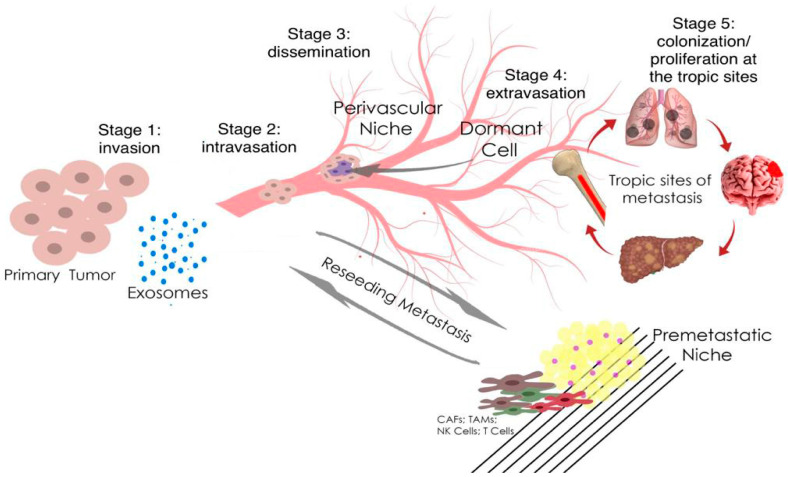
Metastasis and tumor invasion. Tumor cells at the primary tumor sites leave their primary site by dint of invasion into the surrounding stroma (stage 1: invasion). Then they penetrate (stage 2: intravasation) and migrate through the blood or lymph vessels into the circulation (stage 3: dissemination). Some of tumor cells survive and migrate to the target organ parenchyma (tropic sites of metastasis), which is the “pre-metastatic niche” (stage 4: extravasation). Finally, cells adapt and proliferate to form metastasis with the subsequent formation of a clinically obvious tumor (stage 5: colonization/proliferation at the tropic sites). Myeloid cells, such as tumor-associated macrophages (TAMs), can promote metastatic spread through blood and lymphatic vessels. Transforming growth factor-β (TGF-β), produced by TAMs, and cancer-associated fibroblasts (CAFs) are regulators of the epithelial–mesenchymal transition and metastasis. In the pre-metastatic niche, natural killer cells (NK cells) and T cells inhibit tumor growth.

**Figure 2 ijms-24-01980-f002:**
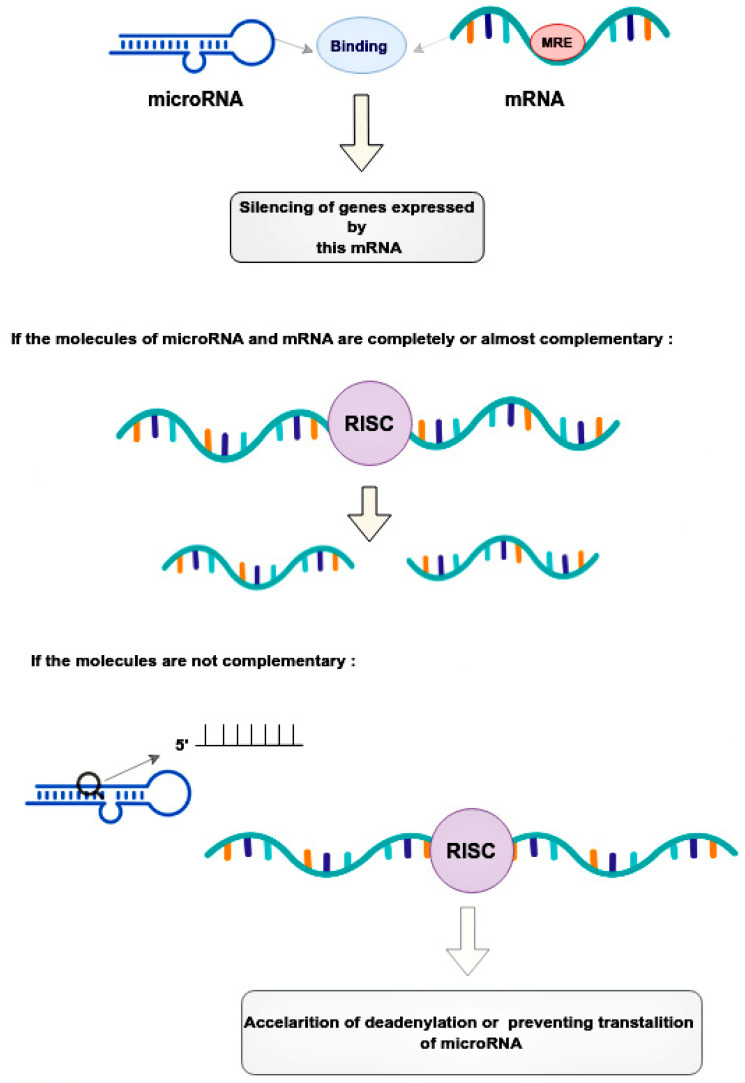
Regulatory patterns of mRNA and microRNA interaction. MicroRNAs bind to the mRNA MRE complex in 3’-untranslated regions, resulting in the post-transcriptional silencing of genes. The mode of silencing in RISC depends whenever mRNA and microRNA are completely or partially complementary. In cases where molecules are perfectly complementary, the RISC simply cleaves the target mRNA. In cases where molecules are no perfectly complementary, the RISC binds to the MRE only by the heptameric 5’seed region of the guide microRNA, leading to acceleration of mRNA decay. This happens due to deadenylation or interruption translation at its different stages. RISC—the RNA-induced silencing complex, MRE—microRNA response element.

**Figure 3 ijms-24-01980-f003:**
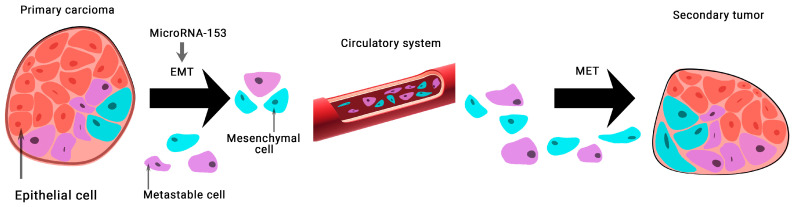
The role of microRNA-153 in carcinogenesis. MicroRNA-153 is involved in epithelial-mesenchymal transition (EMT)-associated signaling pathways that stimulate tumorigenesis, cancer progression, and metastasis. A mesenchymal–epithelial transition (MET) is a reversible biological process that involves the transition from mesenchymal cells to polarized cells called epithelia.

**Figure 4 ijms-24-01980-f004:**
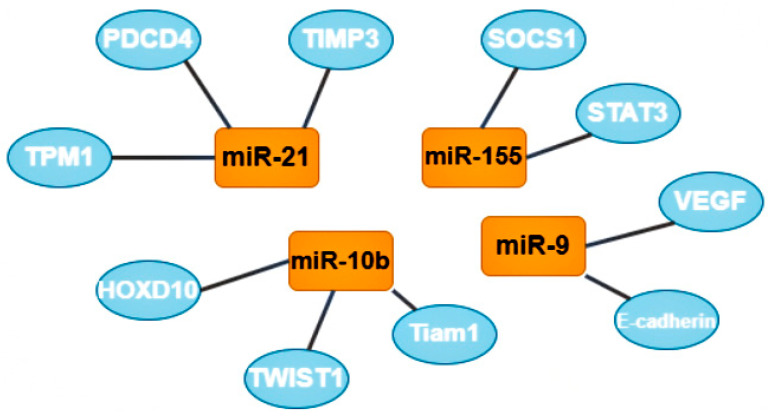
The most studied oncomirs and their target genes. MiR-21 has 3 common target genes: tropomyosin 1 (TPM1), programmed cell death 4 (PDCD4), and metallopeptidase inhibitor 3 (TIMP3). MiR-10b also has 3 target genes, including homeobox D10 (HOXD10), twist-related protein 1 (TWIST1) and T cell lymphoma invasion and metastasis-inducing protein 1 (Tiam 1). Mir-9 target genes are vascular endothelial growth factors (VEGF) and E-cadherin (also known as CDH1). MiR-155 has 2 target genes: suppressor of cytokine signaling 1 (SOCS1) and signal transducer and activator of transcription 3 (STAT3).

**Figure 5 ijms-24-01980-f005:**
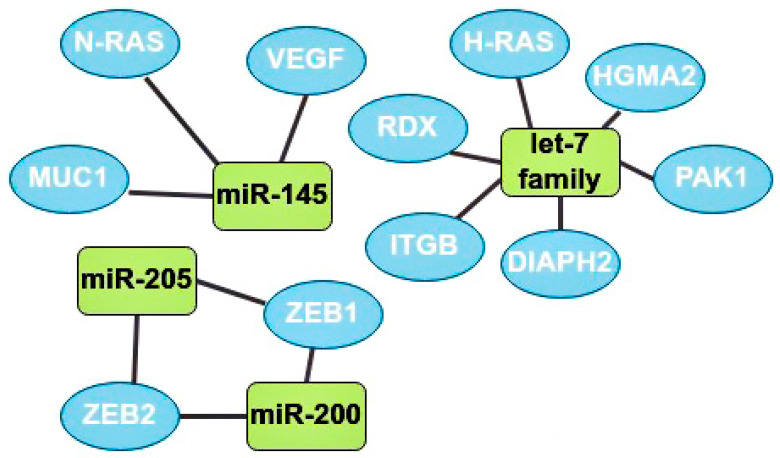
The most studied oncosuppressors and their target genes. MiR-145 has 3 target genes: neuroblastoma RAS viral oncogene homolog (N-RAS), MUC1, and vascular endothelial growth factors (VEGF). Let-7 family target genes: radixin (RDX), Harvey rat sarcoma viral oncogene homolog (H-RAS), high-mobility group AT-hook 2 (HGMA2), serine/threonine-protein kinase (PAK1), diaphanous-related formin 2 (DIAPH2), and integrin subunit beta (ITGB). MiR-205 and miR-200 have 2 mutual target genes: zinc finger E-box binding homeobox 1 and 2 (ZEB1, ZEB2).

**Figure 6 ijms-24-01980-f006:**
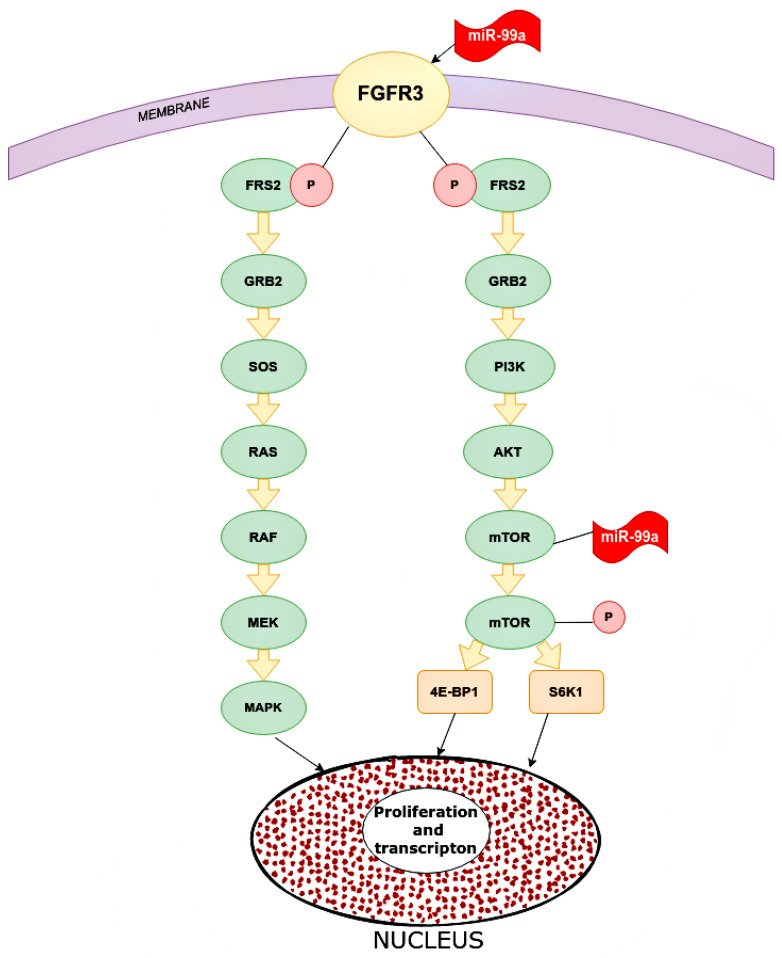
Main pathways in which miR-99a inhibits in cancer cells. microRNA-99a inhibits FGFR3 on the cancer cells. Inhibited FGFR3 cannot activate downstream cascade pathways: PI3K/AKT and RAS/RAF/MEK/MAPK. For the PI3K/AKT signaling pathway, FGFR3 phosphate FRS2 inhibits GRB2. This inhibited molecular PI3K, which inhibits mTOR and as a result, reduces proliferation and transcription in the nucleus. As for the RAS/RAF/MEK/MAPK pathway, inhibited GRB2 inactivate SOS, RAS, RAF, MEK, and MAPK, and eventually increase proliferation and transcription. FGFR2 (CD332)—fibroblast growth factor receptor 2, FRS2—fibroblast growth factor receptor substrate 2, GRB2—growth factor receptor-bound protein 2, PI3K—phosphoinositide 3-kinase, AKT—RAC-alpha serine/threonine-protein kinase, protein kinase B alpha, RAS—rat sarcoma virus kinase, RAF (rapidly accelerated fibrosarcoma)—proto-oncogene serine/threonine-protein kinase, mTOR—mammalian target of rapamycin kinase, MEK (MAP2K, MAPKK)—mitogen-activated protein kinase, MAPK—mitogen-activated protein kinase, 4E-BP1—eukaryotic translation initiation factor 4E-binding protein 1, S6K1—ribosomal protein S6 kinase beta-1.

**Figure 7 ijms-24-01980-f007:**
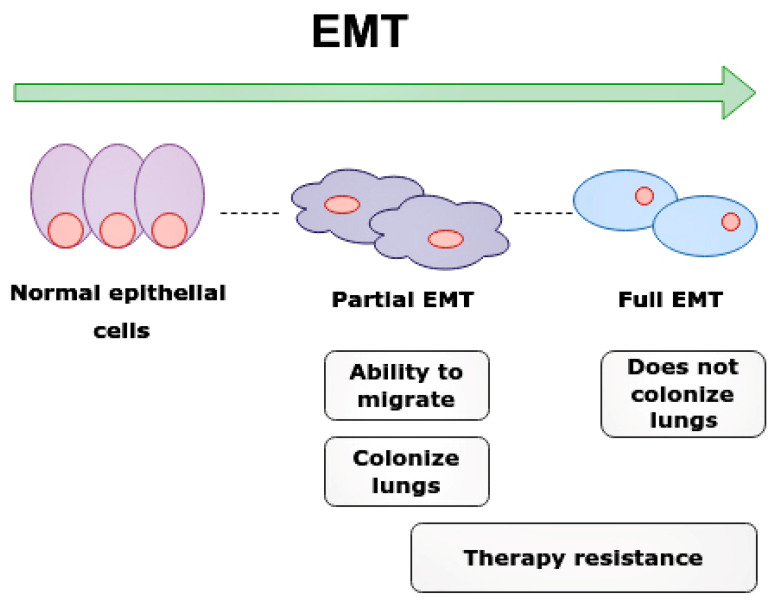
Epithelial-mesenchymal transition (EMT) and clinical significance of partial EMT and full EMT. Cells undergoing partial EMT have an ability to migrate and colonize lungs. Cell in full EMT do not colonize lungs. Therapy resistance is characteristic of both partial EMT and full EMT.

**Figure 8 ijms-24-01980-f008:**
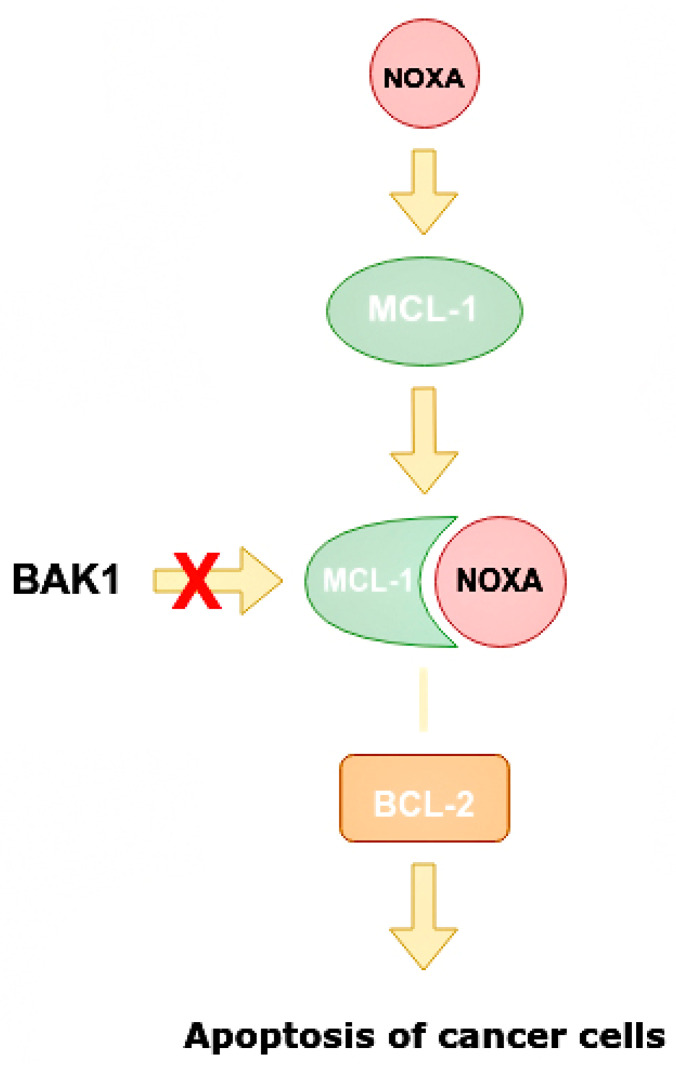
Mechanism of increasing apoptosis of cancer cells when miR-4728 does not inhibit NOXA. NOXA displaces BAK1 from MCL-1 as they have a similar conformation. The MCL-1/NOXA complex activates BCL-2, which activates the apoptosis of cancer cells. NOXA—NADPH oxidase activator, MCL-1—mantle cell lymphoma 1 protein, BCL-1—B cell lymphoma 1 protein, BCL-2—B cell lymphoma 2 protein, BAK1—BCL-2 homologous antagonist killer.

**Figure 9 ijms-24-01980-f009:**
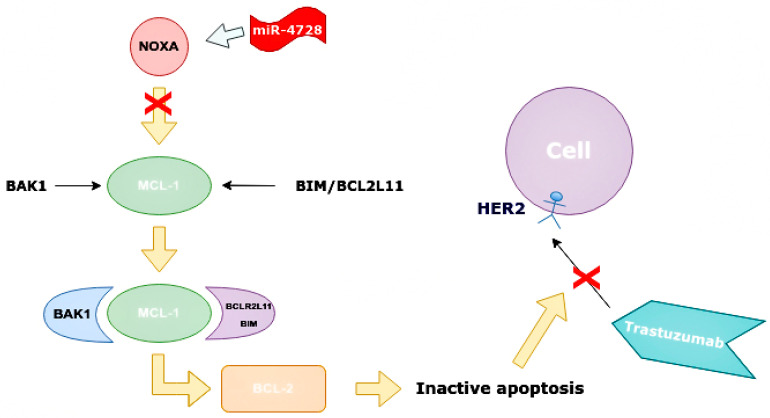
Suppressing effect of miR-4728 in cancer cells. MiR-4728 inhibited NOXA, thus BAK1 and BIM/BCL2L11 can combine to MCL-1. This complex inhibits BCL-2 and inactivates cell apoptosis, therefore, trastuzumab cannot impact on cells with HER2 expression. NOXA—NADPH oxidase activator, MCL-1—mantle cell lymphoma-1, BCL-2—B cell lymphoma 2 protein, BAK1—BCL-2 homologous antagonist killer, HER2—human epidermal growth factor receptor 2, BIM/BCL2L11—Bcl-2 interacting mediator of cell death (commonly called Bcl-2-like protein 11).

**Table 1 ijms-24-01980-t001:** The differences between microRNA and siRNA.

	microRNA	siRNA
Structure	small (18–25 nucleotides) non-coding single-stranded molecules	small (21–23 nucleotides) non-coding double-stranded molecules
Target	Many DNA/mRNA molecules	One mRNA molecule
Sites	Cytoplasm, nucleus	Cytoplasm

**Table 2 ijms-24-01980-t002:** The variety of microRNAs in different subtypes of breast cancer. Cluster 1 contains luminal A samples and some luminal B; Cluster 2 mostly contains luminal B; Cluster 3 includes only basal-like subtype. HER2+ tumors were identified as a sub-cluster as they had features of Cluster 1 and Cluster 2.

	Cluster 1 (ER+)	Cluster 2 (ER−)	Cluster 3 (ER−/PR−/HER2−)	Only for HER2+ Tumors	The Most Common in All Subtypes
microRNA	miR-26, miR-5681a, miR-5695, miR-887, miR-149, miR-375, miR-342, miR-190b, miR-29c, miR-29b miR-499a	miR-455-3p, miR-934, miR-135b miR-577 miR-548ao, miR-584, miR-138, miR-135b, miR-455, miR-577, miR-934	miR-18a miR-516a miR-519a miR-520b miR-522 miR-1283	miR-4728	miR-21-5p

## Data Availability

The data presented in this study are available on request from the corresponding author.

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
