# Peer review of "The Role of Different Types of microRNA in the Pathogenesis of Breast and Prostate Cancer"

_ijms, 2023, doi:10.3390/ijms24031980_

Round 1
Reviewer 1 Report
This is an interesting review on the different types of microRNA and their molecular targets that are involved in the pathogenesis of breast and prostate cancer.
Despite its great interest, some aspects should be improved to increase its reading grade and its impact on the subject.
1. The difference between microRNA and small interfering RNA should be better explained throughout the review. Perhaps the authors can design a table of differences between the two.
2. line 53. BC?
3. The figures should be self-explanatory, indicating the abbreviations, the context of the image and the different observable steps in the image in the figure caption.
4. line 85-97, it is difficult and arduous to understand, perhaps some figure or diagram can benefit and speed up your understanding.
5. line 112. Reference?
6. Fig2, 3, 4 Explain the events in the figure caption.
7. line 141. PTEN?
8.line148. TNM?
9. line 160. Ref?
10. line 164. Be careful with the numbering of the bibliography, here the ref. number 1.
11. Check all the text, there are numerous typos.
12. line 230-252. Focus this part of the text making it more understandable and readable. Perhaps some explanatory table could improve.
13. line 250. Showen?
14. line 275. Ref?
15. line 259-285, Focus this part of the text, making it more understandable and readable. It's hard to read.
16. Fig. 5. The authors could mark the presence of miR-99a in red in the scheme, to improve the visualization of its presence.
17. line 345. Ref?
18. line 351. Figure 6?
19. Fig. 6, 7. Explain the events at the bottom of the figure and the involvement of miRNAs in the different events. Figures should be self-contained.
20. line 447. Figure8? figure? The figure should be placed after citing it in the text.
21. line 446-463. Lack of focus.
22. The authors should reflect on the current limitations of miRNAs.
Author Response
Manuscript ID ijms-2102514
Response to Reviewers'
Thank you for allowing us to submit a revised draft of the manuscript ‘The role of different types of microRNA in the pathogenesis of breast and prostate cancer ’ for publication in the SI "Predictive, Preventive and Personalised (3P) Medicine: From Bench to Bedside" of IJMS MDPI. We appreciate the time and effort you dedicated to providing feedback on our manuscript and are grateful for the insightful comments and valuable improvements to our paper. We have incorporated all of the major corrections. Please see below for a point-by-point response to the reviewer's comments and concerns.
|
Reviewers' Comments |
Authors' Responses |
|
Reviewer 1 This is an interesting review on the different types of microRNA and their molecular targets that are involved in the pathogenesis of breast and prostate cancer. Despite its great interest, some aspects should be improved to increase its reading grade and its impact on the subject.
|
Dear Reviewer 1, Thank you for reviewing our manuscript! We really appreciate your comments and advice! We have corrected the manuscript and proofread the English language. We have corrected some parts of the article that may have been difficult to understand and added images and diagrams for a better understanding of the article. We proofread the text with an English-speaking reviewer to improve the perception of the text by readers.
|
|
1. The difference between microRNA and small interfering RNA should be better explained throughout the review. Perhaps the authors can design a table of differences between the two.
|
We have added explanations in the text and supplemented it with a table to help improve understanding of the review. |
|
2. line 53. BC?
|
We have added an abbreviation for breast cancer (BC).
|
|
3. The figures should be self-explanatory, indicating the abbreviations, the context of the image and the different observable steps in the image in the figure caption. |
Thanks for your comment! We have added descriptions to the figures and deciphered the abbreviations. |
|
4. line 85-97, it is difficult and arduous to understand, perhaps some figure or diagram can benefit and speed up your understanding. |
We have added explanations in the text and supplemented it with a figure to help improve understanding of the review. |
|
5. line 112. Reference?
|
Reference added. |
|
6. Fig2, 3, 4 Explain the events in the figure caption.
|
Thanks for your comment! We have added descriptions to the figures and deciphered the abbreviations. |
|
7. line 141. PTEN? |
Thank you! Changes have been made. |
|
8.line148. TNM? |
Thank you! Changes have been made. |
|
9. line 160. Ref? |
Reference added. |
|
10. line 164. Be careful with the numbering of the bibliography, here the ref. number 1 |
Thank you for your comment! In this sentence we have referred again to the first reference. |
|
11. Check all the text, there are numerous typos.
|
Thank you for your comment! The typos have been corrected. |
|
12. line 230-252. Focus this part of the text making it more understandable and readable. Perhaps some explanatory table could improve.
|
We have added explanations in the text and supplemented it with a table to help improve understanding of the review. |
|
13. line 250. Showen?
|
Thanks for your comment! The typo has been corrected.
|
|
14. line 275. Ref?
|
Reference added. |
|
15. line 259-285, Focus this part of the text, making it more understandable and readable. It's hard to read.
|
The paragraph has been rewritten. |
|
16. Fig. 5. The authors could mark the presence of miR-99a in red in the scheme, to improve the visualization of its presence.
|
Agree with comment. Changes in the figure 5 have been made. |
|
17. line 345. Ref?
|
Reference added. |
|
18. line 351. Figure 6?
|
Figure 6 moved throughout the text. |
|
19. Fig. 6, 7. Explain the events at the bottom of the figure and the involvement of miRNAs in the different events. Figures should be self-contained.
|
Thanks for your comment! We have added descriptions to the figures and deciphered the abbreviations. |
|
20. line 447. Figure8? figure? The figure should be placed after citing it in the text.
|
Figure 8 moved throughout the text. |
|
21. line 446-463. Lack of focus.
|
The paragraph has been rewritten. |
|
22. The authors should reflect on the current limitations of miRNAs.
|
Thanks for your comment! We have described the existing limitations in the conclusion. |

Reviewer 2 Report
The Review aims to summarize the current information on the role of micro ribonucleic acids (microRNAs, miRNAs) in prostate and breast cancers development, an opportunity to use microRNAs as prognostic biomarkers for different types of cancer which might help to enhance patients survival.
Authors provided schemes, which help to understand information.
In the Review there is a paragraph, which is devoted to the role of miRNAs in prostate cancer, probably it would be logical to have the paragraph, the title of which would include the name of the other type of cancer which Authors study? But it is very much upon the Authors point of view, only if they would consider that it would be more appropriate.
It would be good in the Conclusion to summarize the information in a connection with those types of cancers, which are studied.
The following comments do not diminish the value of the Review:
Line 53 Probably it would be better to first use the abbreviature BC near the term breast cancer (line 52).
Line 55, 56 Does the number of cases and diseases refer to the whole population in the world?
Line 74 Probably it would be good to add an explanation to the Figure 1. What does ‘Tropic sites of metastasis’ mean?
Line 111 In the paragraph 2. 'The role miRNA in prostate cancer' there is also an information about miRNA expression in breast cancer, probably it would be better to rename the paragraph?
Line 128 The following phrase should be checked ‘It was founded’.
Line 136 Probably it would be better to add an explanation to the Figure 2, including description of the abbreviations EMT, MET.
Line 138 Does the following phrase ‘was founded in 2013’ mean discovered?
Line 141 Abbreviation PTEN should be explained.
Line 148 TNM abbreviation should be explained.
Line 162 The part of the sentence ‘metastasis[27]’ should be changed to the ‘metastasis [27]’.
Line 168 The word ‘oncomire’ should be changed to ‘oncomir’.
Line 182 The part of the sentence ‘transition[1]’ should be replaced with ‘transition [1]’.
Line 183 The word ‘oncomiras’ should be replaced with ‘oncomires’.
Line 186 The part of the sentence ‘(Figure3 )[27].’ Should be replaced with ‘(Figure3) [27].’.
Line 193 Probably it would be quite informative to add details about the function of mentioned genes.
Line 203 The part of the sentence ‘self-renewal[30]’ should be replaced with ‘self-renewal [30]’.
Line 205 The part of the sentence ‘ITGB[1]’ should be replaced with ‘ITGB[1]’.
Line 211, 212 The part of the sentence ‘tumors[31]’ should be replaced with ‘tumors [31]’.
Line 218 To the part of the sentence ‘nodes[32]’ the space should be added ‘nodes [32]’.
Line 220 To the part of the sentence ‘miR-141.Most’ the space should be added ‘miR-141. Most’.
Line 228 Probably it would be quite informative to add details about the function of mentioned genes.
Line 230 Probably it would be better to start a new paragraph at that point?
Line 233 The word ‘и’ should be replaced with ‘and’.
Line 234 ‘[34].Technical’ space should be added ‘[34]. Technical’.
Line 243 Would you please explain, what is the meaning of the phrase ‘the most common being’?
Line 262 The part of the sentence ‘receptor 3)’ should be replaced with ‘receptor 3).’
Line 266 The following word ‘mediaotors’ should be replaced with ‘mediators’.
Line 268 The part of the sentence ‘(Figure 5.)’ would be better to replace with ‘(Figure 5)’.
Line 270 What would be the description of the abbreviature ‘PI3P/AKT’, would be better if it would be described.
It would be better to include the descriptions of all the mentioned in the text abbreviations.
Line 318 Probably it would be better to replace ‘NUCLEO’ with ‘nucleus’, ‘Proleferation’ should be replaced with ‘Proliferation’.
Lines 351, 352, 354, 362, 364 ‘E-cateine’ has to be checked.
Line 353 The part of the sentence ‘transmission[ 46].’ Would be better to replace with ‘transmission [46].’
Line 358 Does the following word ‘metastas’ mean ‘metastasize’?
Line 373 Probably it would be better to add descriptions to the abbreviations used in Figure 6.
Line 378 Probably the word ‘bings’ should be replaced with ‘binds’?
Line 395 The word ‘amlification’ should be checked.
Line 399 The font of title should be bold, as the other titles through the text.
Line 415 The phrase ‘on MCL1 based on similarity’ should be checked.
Line 416 The part of the sentence ‘MCL1[49].’ should be replaced with ‘MCL1 [49].’
References should be corrected according the recommendations, published on the Journal website.
Author Response
Thank you for allowing us to submit a revised draft of the manuscript ‘The role of different types of microRNA in the pathogenesis of breast and prostate cancer ’ for publication in the SI "Predictive, Preventive and Personalised (3P) Medicine: From Bench to Bedside" of IJMS MDPI. We appreciate the time and effort you dedicated to providing feedback on our manuscript and are grateful for the insightful comments and valuable improvements to our paper. We have incorporated all of the major corrections. Please see below for a point-by-point response to the reviewer's comments and concerns.
|
Reviewers' Comments |
Authors' Responses |
|
Reviewer 2 The Review aims to summarize the current information on the role of micro ribonucleic acids (microRNAs, miRNAs) in prostate and breast cancers development, an opportunity to use microRNAs as prognostic biomarkers for different types of cancer which might help to enhance patients survival. Authors provided schemes, which help to understand information. In the Review there is a paragraph, which is devoted to the role of miRNAs in prostate cancer, probably it would be logical to have the paragraph, the title of which would include the name of the other type of cancer which Authors study? But it is very much upon the Authors point of view, only if they would consider that it would be more appropriate. It would be good in the Conclusion to summarize the information in a connection with those types of cancers, which are studied.
|
Dear Reviewer 2, Thank you for reviewing our manuscript! We really appreciate your comments and advice! We have corrected the manuscript and proofread the English language. Thanks for your comment! Agree with comment. We have added a paragraph indicating breast cancer. We accepted your comment and summarized the information in the conclusion of the manuscript. |
|
Line 53 Probably it would be better to first use the abbreviature BC near the term breast cancer (line 52). |
We have added an abbreviation for breast cancer (BC).
|
|
Line 55, 56 Does the number of cases and diseases refer to the whole population in the world? |
Yes, the numbers of diseases are given for the world's population.Сhanges have been made.
|
|
Line 74 Probably it would be good to add an explanation to the Figure 1. What does ‘Tropic sites of metastasis’ mean? |
Thanks for your comment! We have added descriptions to the figures and deciphered the abbreviations. |
|
Line 111 In the paragraph 2. 'The role miRNA in prostate cancer' there is also an information about miRNA expression in breast cancer, probably it would be better to rename the paragraph? |
Thanks for your comment, changes have been made. |
|
Line 128 The following phrase should be checked ‘It was founded’. |
Changes applied. It was founded that miRNA-153 expression in breast cancer tissue samples and MDA-MB-231 cells was significantly lower than normal. à The expression of miRNA-153 in breast cancer tissue samples and 128 MDA-MB-231 cells was found to be significantly lower than normal. |
|
Line 136 Probably it would be better to add an explanation to the Figure 2, including description of the abbreviations EMT, MET. |
Thanks for your comment! We have added descriptions to the figures and deciphered the abbreviations. |
|
Line 138 Does the following phrase ‘was founded in 2013’ mean discovered? |
Thanks for your comment! The typo has been corrected. |
|
Line 141 Abbreviation PTEN should be explained. |
Thanks for your comment, changes have been made. |
|
Line 148 TNM abbreviation should be explained. |
Thanks for your comment, changes have been made. |
|
Line 162 The part of the sentence ‘metastasis[27]’ should be changed to the ‘metastasis [27]’. |
Changes applied. |
|
Line 168 The word ‘oncomire’ should be changed to ‘oncomir’. |
Thanks for your comment! The typo has been corrected. |
|
Line 182 The part of the sentence ‘transition[1]’ should be replaced with ‘transition [1]’. |
Changes applied. |
|
Line 183 The word ‘oncomiras’ should be replaced with ‘oncomires’. |
Thanks for your comment! The typo has been corrected. |
|
Line 186 The part of the sentence ‘(Figure3 )[27].’ Should be replaced with ‘(Figure3) [27].’. |
Changes applied. |
|
Line 193 Probably it would be quite informative to add details about the function of mentioned genes. |
Thanks for your comment! We have added descriptions to the figures. |
|
Line 203 The part of the sentence ‘self-renewal[30]’ should be replaced with ‘self-renewal [30]’. |
Changes applied. |
|
Line 205 The part of the sentence ‘ITGB[1]’ should be replaced with ‘ITGB[1]’. |
Changes applied. |
|
Line 211, 212 The part of the sentence ‘tumors[31]’ should be replaced with ‘tumors [31]’. |
Changes applied. |
|
Line 218 To the part of the sentence ‘nodes[32]’ the space should be added ‘nodes [32]’. |
Changes applied. |
|
Line 220 To the part of the sentence ‘miR-141.Most’ the space should be added ‘miR-141. Most’. |
Changes applied. |
|
Line 228 Probably it would be quite informative to add details about the function of mentioned genes. |
Thanks for your comment! We have added descriptions to the figures. |
|
Line 230 Probably it would be better to start a new paragraph at that point? |
Thank you! Changes have been made. |
|
Line 233 The word ‘и’ should be replaced with ‘and’. |
Thanks for your comment! The typo has been corrected. |
|
Line 234 ‘[34].Technical’ space should be added ‘[34]. Technical’. |
Changes applied. |
|
Line 243 Would you please explain, what is the meaning of the phrase ‘the most common being’? |
Thanks for your comment! The typo has been corrected. |
|
Line 262 The part of the sentence ‘receptor 3)’ should be replaced with ‘receptor 3).’ |
Changes applied. |
|
Line 266 The following word ‘mediaotors’ should be replaced with ‘mediators’. |
Thanks for your comment! The typo has been corrected. |
|
Line 268 The part of the sentence ‘(Figure 5.)’ would be better to replace with ‘(Figure 5)’. |
Changes applied. |
|
Line 270 What would be the description of the abbreviature ‘PI3P/AKT’, would be better if it would be described. It would be better to include the descriptions of all the mentioned in the text abbreviations. |
Thanks for your comment! Explained earlier in the text where the abbreviation was encountered. |
|
Line 318 Probably it would be better to replace ‘NUCLEO’ with ‘nucleus’, ‘Proleferation’ should be replaced with ‘Proliferation’. |
Thanks for your comment! The typo has been corrected. |
|
Lines 351, 352, 354, 362, 364 ‘E-cateine’ has to be checked. |
Thanks for your comment! The typo has been corrected. |
|
Line 353 The part of the sentence ‘transmission[ 46].’ Would be better to replace with ‘transmission [46].’ |
Changes applied. |
|
Line 358 Does the following word ‘metastas’ mean ‘metastasize’? |
Thanks for your comment! The typo has been corrected. |
|
Line 373 Probably it would be better to add descriptions to the abbreviations used in Figure 6. |
Thanks for your comment! We have added descriptions to the figures. |
|
Line 378 Probably the word ‘bings’ should be replaced with ‘binds’? |
Thanks for your comment! The typo has been corrected. |
|
Line 395 The word ‘amlification’ should be checked. |
Thanks for your comment! The typo has been corrected. |
|
Line 399 The font of title should be bold, as the other titles through the text. |
Changes applied. |
|
Line 415 The phrase ‘on MCL1 based on similarity’ should be checked. |
Thank you! All changes have been made. |
|
Line 416 The part of the sentence ‘MCL1[49].’ should be replaced with ‘MCL1 [49].’ |
Changes applied. |
|
References should be corrected according the recommendations, published on the Journal website. |
Thank you! Changes applied. |

Round 2
Reviewer 1 Report
All abbreviations included in figures should be indicated in their legend.
Figure 1 could list the different stages
Author Response
Manuscript ID ijms-2102514
Response to Reviewers'
Thank you for allowing us to submit a revised draft of the manuscript ‘The role of different types of microRNA in the pathogenesis of breast and prostate cancer ’ for publication in the SI "Predictive, Preventive and Personalised (3P) Medicine: From Bench to Bedside" of IJMS MDPI. We appreciate the time and effort you dedicated to providing feedback on our manuscript and are grateful for the insightful comments and valuable improvements to our paper. We have incorporated all of the minor corrections. Please see below for a point-by-point response to the reviewer's comments and concerns.
|
Reviewers' Comments |
Authors' Responses |
|
Reviewer 1 All abbreviations included in figures should be indicated in their legend.
|
Dear Reviewer 1, Thank you for reviewing our manuscript! All abbreviations have been added.
|
|
Figure 1 could list the different stages |
Thank you for your comment! In Figure 1 the stages and an extended description in the legend have been added. |
